# The Effect of Follower Identity on Followership: The Mediating Role of Self-Efficacy

**DOI:** 10.3390/bs13060482

**Published:** 2023-06-07

**Authors:** Weixi Zeng, Zheming Xu, Lixia Zhao

**Affiliations:** 1Yangtze Delta Region Institute (Huzhou), University of Electronic Science and Technology of China, Huzhou 313001, China; 2School of Public Administration, University of Electronic Science and Technology of China, Chengdu 611731, China

**Keywords:** followership, follower identity, followership prototypes, self-efficacy

## Abstract

Followership is as crucial as leadership for organizational success. Significant efforts have been made by numerous researchers to examine how leadership influences followership; however, not enough attention has been paid to the influence of internal factors of followers on followership from the followers’ perspective. This study relies on identity theory to understand the relationship between the influence of followers’ perceived self-following traits (FTP) and followership prototype (FP) on followership, and the mediation role of self-efficacy in the relationship between FTP-FP consistency and followership. In order to avoid common method bias and ensure good discriminant validity of the variables, a two-wave time-lagged data collection design was used to collect 276 valid questionnaires from front-line business staff and junior supervisors in private and public sector organizations of China. Polynomial regression and response surface analysis were used to investigate the effect of FTP-FP consistency on followership. The empirical findings indicated that (1) the more consistent FTP-FP, the stronger the followership; (2) compared to the ‘low FTP-low FP’, employees with ‘high FTP-high FP’ had stronger followership; (3) employees with ‘high FTP-low FP’ had stronger followership than ‘low FTP-high FP’; (4) self-efficacy played a mediating role between FTP-FP consistency and followership. These findings contribute to management practice by revealing antecedents to followership from the perspective follower identity and the effect of follower identity on followership.

## 1. Introduction

As society evolves rapidly and organizations face increasingly complex external challenges, teamwork gradually replaces the leader’s charisma as a critical factor for organizational success [1,2,3]. The importance of followership as a series of proactive and cooperative behaviors that employees exhibit in following the leader [4,5] is becoming more and more prominent. The employees’ followership positively affects both leadership and organizational performance. For example, followers who are too passive and submissive are more likely to accept unethical demands made by the leader and develop a blind authority cult [6]. In contrast, active followers who exhibit “co-production” are more likely to express themselves proactively and contribute to achieving higher organizational performance [7]; therefore, understanding the nature and influencing factors of followership is essential for organizational HRM optimization and success.

Although the influence of followership in organizations has received much attention and scholars have explored the causes of followership from multiple perspectives, most of these studies tend to focus on the leader, mainly exploring the influence of the leader’s implicit cognition, leadership behavior, leadership style, and other factors on followership [8,9,10]. Significant efforts have been made by numerous researchers to examine how transformational leadership [3,8,11], workplace authentic leadership [12], ethical leadership [6,13], servant leadership [14] and humble leadership [15] affects followership or other employees’ behaviors.

However, not enough attention has been paid to the influence of internal factors of followers on followership from the followers’ perspective [16,17]. It has been noted that it is difficult for leaders to change their followers’ thoughts, feelings, and values [18]. The internal characteristics of followers, including role orientation, motivation to follow, cognitive ability, and relational skills, are essential factors that influence followership [4]. Follower identity, an employee’s self-social role orientation in the workplace, provides employees with a psychological framework and guidelines for action in the workplace and has a significant potential impact on followership [9]. The causes of followership need to be studied from a more diverse perspective, especially from the internal perspective of the followers [7]. 

According to Epitropaki et al. [19], we should take stock of information-processing approaches to leadership and followership in organizational settings. A leader identity approach was used to understand the motivation to lead [20]; it was found that projective identification affects leader’s leadership [21]. Since a leader’s identity influences his or her leadership, it is likely that followership is also influenced by the identity of the followers; however, little research exists on the direct link between followers’ identity and their followership. This study was grounded in the prediction of a follower identity approach to address the above-mentioned existing research gaps [4].

Furthermore, enough attention has been given to the intervening mechanism that exists between leadership and employee behaviors [3,22], numerous researchers examined the mediation role of self-efficacy between leadership and employee engagement [13], followers’ organizational citizenship behavior [14], and employees’ knowledge-hiding behavior [15]. To the best of our knowledge, only Xiong and Epitropaki discussed a direct link between followers’ identity and followership [17,19], while no other study has examined the role of self-efficacy in a follower as mediators between follower identification and followership. So, the gap that exists in understanding the intervening mechanism that is present between followers’ identification and followership needs to be bridged; therefore, this paper will explore the influence of follower identity and self-efficacy on employees’ followership from the perspective of the followers.

This study develops theoretical deductions and empirical analyses around three core concepts: followership prototype (FP), self-followership traits perception (FTP), and self-efficacy (SE). The followership prototype is derived from implicit followership theories (IFTs), which are individuals’ cognitive schemas about the traits and behaviors of influential followers. Based on implicit followership theories, the followership prototype in this study refers to the implicit stereotypes that followers have about good followers, i.e., what traits good followers should possess. It reflects the ideal state of followership traits in the minds of followers. Self-following trait perception (FTP) refers to the follower’s perception and evaluation of the reality of their following traits, which is the follower’s perception of the actual state of their following traits. This paper will explore whether employees with consistent FTP-FP have stronger followership than those with inconsistent FTP-FP. Do employees with high FTP-low FP have higher followership than those with low FTP-high FP? How does FTP-FP congruence affect followers’ self-efficacy and, thus, their followership? Ultimately, this study will fill the gap of knowledge that exists between follower identity, self-efficacy and followership, and make a significant contribution to the current literature in a variety of ways.

## 2. Theoretical Basis and Research Hypothesis

### 2.1. Identity Theory

Identity theory concerns the psychological and behavioral processes by which individuals play social roles [23]. Identity is an individual’s self-categorization and self-definition at the level of social roles, which provides a psychological framework for interpreting social situations and personal behavior and influences the individual’s cognitive, emotional, and behavioral performance [24]. The establishment of identity depends on identity construction or identity work, in which individuals construct, modify, and refine their social roles through a series of psychological operations to establish an identity [25,26,27]. In identity construction, the identity motive plays a critical guiding role [26]. Among them, self-verification, as a vital identity motive, drives individuals to compare their perceived self-performance with the identity standard (positive or negative) to maintain the stability of their existing identity. The identity standards used in the self-validation process are also known as identity prototypes, representing the individual’s central characteristics and idealized state of a social role [24]. The followership prototype (FP), derived from implicit followership theories (IFTs), is an individual’s cognitive schema of influential followers’ traits and behavioral patterns. According to the description of this theory, influential followers possess characteristics such as diligence, enthusiasm, and loyalty [27]. These schemas about follower traits are formed during socialization and stored in individuals’ memories, and individuals use them as identity criteria to form judgments and evaluations of followers. In particular, the follower prototype (FP) includes both the leader’s assumptions about the characteristics of the followership of subordinates and the subordinates’ expectations about the traits of their own followership. From the follower’s perspective, the follower prototype (FP) is an identity reference for employees when they play the role of a follower [9]. Employees can not only rely on the followership prototype to form idealized role expectations about themselves but also externalize the followership prototype to perceive and evaluate their actual performance. In this study, we will refer to past research methods to indirectly measure the identity of followers through the matching consistency of the two, using the following prototype (FP) with schematic features as an identity criterion and the perceived self-following trait (FTP) as an externalized performance [27].

### 2.2. The Effect of FTP-FP Consistency on Followership

FP is an individual’s self-identification with the follower role, a self-perception about being a follower, and how one sees oneself as a follower. When FTP and FP are aligned, self-validation is achieved, and follower identity in the workplace is clear. Follower identity can generate stronger motivation to follow and trigger a range of positive, proactive follower behaviors, e.g., more constructive behaviors, more substantial organizational commitment, and execution [28,29,30]. On the contrary, if FTP fails to match with FP, employees will experience cognitive dissonance, experience ego conflicts [31] and have difficulty establishing an identity. According to self-discrepancy theory, when there is a gap between the actual self and the ideal self, individuals will develop negative emotions, such as frustration, disappointment, and dissatisfaction, because they do not meet the self-standard [32], leading to a lower level of self-confidence and self-esteem [33], which prevents individuals from successfully playing the follower role and in turn weakens their followership. Accordingly, the following hypothesis is proposed.

**Hypothesis** **1** **(H1).**
*The higher the alignment between FTP and FP, the stronger the employee’s followership.*


In the case of FTP and FP congruence, different levels of congruent matching may also impact employee followership. Identity theory suggests that the activation of identity initiates a self-validation mechanism in which individuals actively attend to seek out information that is consistent with the role, behave in a manner consistent with identity standards, and strive to maintain their outward performance as close to the prototype or identity standards as possible [24]. It is inferred that with consistent FTP-FP, followers will exhibit behaviors consistent with their following prototypes. Followers with “high FTP-high FP” exhibit more positive followership traits in the work environment, such as being passionate and productive at work, enthusiastic, loyal, and reliable to colleagues and supervisors [34,35]. In contrast, “low FTP-low FP” followers exhibit fewer positive followership traits, which confirms their “low standard”. In the inconsistent situations of “high FTP-low FP” or “low FTP-high FP”, individuals will continuously correct their previous role judgments and choices to rationalize their following behaviors [24,27]. This process consumes a large number of cognitive resources, thus weakening their motivation to follow; however, “low FTP-low FP” followers do not have a similar cognitive resource consumption process, and such followers should have higher followership power than individuals with inconsistent identities. Accordingly, the following hypotheses are proposed.

**Hypothesis** **2** **(H2).**
*In the case of consistency, “high FTP-high FP” individuals have stronger followership than “low FTP-low FP” individuals.*


In playing the follower role, “low FTP-high FP” employees’ evaluation of their external followership traits is far below the ideal standard. It is challenging to meet their self-expectations, so they will experience more frustration and lower self-esteem [36], which weakens their motivation to follow and negatively affects followers’ behavior [29]. In contrast, although employees with “high FTP-low FP” fail to achieve identity, they perceive that external performance exceeds their expectations, which can compensate, to some extent, for the negative emotions caused by the inconsistency between FTP and FP; however, this self-efficacy brought by exceeding expectations needs to be improved. Due to the ego gap, employees must establish a follower identity fully and have difficulty stimulating strong motivation to follow. Accordingly, the following hypothesis is proposed.

**Hypothesis** **3** **(H3).**
*In the case of inconsistency, “high FTP-low FP” has more substantial followership than “low FTP-high FP” individuals.*


### 2.3. The Mediating Role of Self-Efficacy

Self-efficacy (SE) refers to a person’s belief that he or she can successfully and effectively perform tasks in the face of different work tasks and work environments [30,37]. Success and failure experiences, emotional and physical states, demonstration effects, and social persuasion are the primary sources of information about self-efficacy. When FTP is consistent with FP, follower identity is established. Employees can repeatedly confirm the consistency of external performance with identity criteria under the self-validation mechanism, acquire successful experiences of playing the follower role, experience more positive emotions, increase self-confidence, and build strong self-efficacy [38]. When FTP is not aligned with FP, employees not only repeatedly experience the failure of being unable to establish identity but also experience ambivalence and anxiety brought about by cognitive dissonance and self-differences, thus reducing their self-efficacy.

Self-efficacy influences employees’ discrepancy-reduction strategies [39]. In order to cut down negative experiences caused by the difference between actual performance and expected standards, individuals with low self-efficacy will focus on the defects of the self and avoid the goal by lowering the expected standard or even directly choosing to withdraw and abandon the task. In contrast, individuals with high self-efficacy will adopt achievement convergence strategies and achieve the expected goal by putting more effort into it. High self-efficacy is not only effective in enhancing employees’ sense of control over their work so that they can persevere and overcome various difficulties in their work [40] but also in improving their level of work engagement [41], performance, etc. [42].

**Hypothesis** **4** **(H4).***Self-efficacy mediates between FTP-FP consistency and followership*.

The hypothetical model for this study is shown in Figure 1.

## 3. Research Methodology

### 3.1. Study Sample and Procedures

The data for this study came from four government departments and three private companies sourced from Sichuan and Guangdong provinces, covering the financial, medical, and service industries. In China, without a reasonable identity, it is difficult to gain the cooperation of research subjects when entering enterprises and institutions to conduct research. The researcher in this study conducted a convenient sampling with the help of corporate trainers’ identities. The human resources department supported and cooperated with this study, and the participants were identified and coded in advance with the department heads before the study. In order to ensure the quality of the data and to take into account the patience of the research respondents in answering the questions, we opted for a two-wave time-lagged design to collect data. A total of 400 grassroots employees (including front-line business staff and junior supervisors) were invited to participate in a structured questionnaire. Since the same questionnaire was used to measure both external and implicit followership traits and to avoid homogenous variation and confounding effects from using the same method, the researchers divided the questionnaires into two sets, A and B, and conducted two paired surveys. Questionnaire A investigated followership prototype (FP) and self-efficacy (SE), and questionnaire B measured perceived self-following trait (FTP) and followership. Questionnaires A was administered in a group setting through an online test tool during the centralized training sessions in each department; questionnaire B was distributed to employees who gave valid responses in the first survey by mail one month later. A total of 380 sets of questionnaires were distributed in the survey. After collecting and matching the A and B questionnaires and eliminating invalid questionnaires, 276 valid questionnaires were finally screened out, with a recovery efficiency of 72.63%. The proportion of males in the sample was 49.10%; in total, 63.80% had a bachelor’s degree or higher, 56.80% had more than five years of experience, and 58.40% were over 25 years old.

### 3.2. Measurement Tools

FP and FTP measurements were scored on a 9-point Likert scale ranging from “1” for “strongly disagree” to “9” for “strongly agree. “The other scales were scored on a 7-point Likert scale ranging from “1” for “strongly disagree” to “7” for “strongly agree. “The Chinese scales were translated from the English version, and English master’s students were invited to do the “translation-back translation” to ensure the accuracy of the scales.

The FP was measured using the implicit followership scale developed by Sy [34], in which nine items involving positive prototypes were selected, including three dimensions of diligence, enthusiasm, and good citizenship, such as “loyal” and “passionate”. The guide asked subjects to evaluate how each description matched their ideal employee characteristics. The Cronbach’s alpha coefficient for the scale was 0.94.

The FTP was measured regarding Epitropaki and Martin’s method [43], also using the implicit followership scale described above, except that in the guiding phrase, subjects were asked to rate the extent to which each word matched their actual performance. The Cronbach’s alpha coefficient for the scale was 0.92.

Self-efficacy (SE) was measured using Spreitzer’s self-efficacy subscale of the psychological empowerment scale [44]. The scale consists of three items, including “I feel confident in my performance”. The scale’s Cronbach’s alpha coefficient was 0.89.

The followership scale developed by Wenjie Zhou et al. was used to measure followership [5]. The scale consists of 21 questions, including “respectful learning”, “understanding intentions”, “active implementation”, “loyalty and dedication”, “effective communication”, and “authority maintenance”. The scale has six dimensions: “Respectful Learning”, “Understanding Intentions”, “Active Implementation”, “Loyalty and Commitment”, “Effective Communication”, and “Assertion of Authority”. An example is “I will not openly disagree with the leader”. The guideline asks subjects to choose the option that matches the question’s description based on their actual behavior. The Cronbach’s alpha coefficient for the scale was 0.96.

Control variables: this study controlled for demographic variables such as gender, age, length of service, type of organization, and literacy [20].

### 3.3. Data Analysis Techniques

Polynomial regression (PR) and response surface analysis (RSA) were used to investigate the effect of FTP-FP consistency on followership. First, a polynomial regression equation *Y* = *b*_0_ + *bX*_11_ + *bX*_22_ + *b X*_31_^2^ + *b*_4_
*XX*_12_ + *b X*_52_^2^ + *e* is established, where *b*_0_ is a constant; *b*_1_, *b*_2_, *b*_3_, *b*_4_ are regression coefficients; *e* is the error; *Y represents the followership*; *X*_1_ represents FTP; *X*_2_ represents FP and *XX*_12_; and *X*_1_^2^, and *X*_2_^2^ represent the product terms of the predictor variables and their respective squares, respectively. All variables are decentered. Subsequently, the coefficients were obtained by polynomial regression analysis, and the curvature (*b*_3_ − *b*_4_ + *b*_5_) of the response surface along the corresponding cross-section of the non-consistent line (*Y* = −*X*) was calculated to test H1, and the slope (*b*_1_ − *b*_2_) to test H3. The slope (*b*_1_ + *b*_2_) of the response surface along the corresponding cross-section of this consistent line (*Y* = *X*) was calculated to test H2.

Before the mediating effect test, the predictor variables are first multiplied by the corresponding regression coefficients and summed to obtain the new block variable. Then, the effect sizes of the independent variable on the mediating variable *a*_1_, the effect size of the mediating variable on the dependent variable *a*_2_, and the direct effect of the independent variable on the dependent variable after controlling for the mediating variable *a_3_* were calculated separately. Finally, the significance of *a*_1_
*× a*_2_ was calculated to test the mediating effect, and *a*_1_ × *a*_2_ × *a*_3_ was calculated to determine the direction of the omitted mediator. The mediating effect values were estimated by the Bootstrap method with 10,000 sampling at a 95% confidence interval.

## 4. Results Analysis

Factor analysis was performed using AMOS 22.0 to test the discriminant validity. A common method variance test was performed using SPSS 22.0. The coefficients of the polynomial regression equation were estimated and tested, and mediating effect test was performed. Origin 2018 was used to plot the response surface 3D graph.

### 4.1. Common Method Variance Test

We used the Harman one-way method to test common method variance. Factor analysis resulted in a KMO value of 0.94, a Bartlett’s sphericity test χ^2^ of 11,839.27, and a degree of freedom of 1035. Exploratory factor analysis had nine factors. In the unrotated case, the first factor explained 41.25% of the variance, and the nine factors together explained 76.56% of the variance. There were no factors with exceptionally high explanatory power, so there was no common method bias.

### 4.2. Discriminant Validity Test

The confirmatory factor analysis was used to test the discriminant validity of all variables. As shown in Table 1, the hypothesized four-factor model had the best fit compared to other competing models, indicating good discriminant validity of the variables.

### 4.3. Descriptive Statistics

As shown in Table 2, correlations between variables were significant. FTP was significantly and positively correlated with FP (*r* = 0.26, *p* < 0.01), self-efficacy (*r* = 0.63, *p* < 0.01), and followership (*r* = 0.57, *p* < 0.01). FP was significantly and positively correlated with self-efficacy (*r* = 0.43, *p* < 0.01) and followership (*r* = 0.45, *p* < 0.01). Self-efficacy was significantly and positively correlated with followership (*r* = 0.72, *p* < 0.01).

### 4.4. Hypothesis Testing

#### 4.4.1. Response Surface Analysis

Polynomial regression with response surface analysis was used to examine the effect of FTP-FP consistency on followership. Predictor variable scores were first standardized and bounded by ±0.5 standard scores, and the proportion of paired samples was counted. The results showed that 22.83% of the subjects had an FTP greater than FP, and 21.38% of the subjects had an FTP less than FP, with nearly half of the sample in discordant pairs, indicating that the sample was suitable for response surface analysis [45]. Polynomial regression analysis (Table 3) and response surface plots (Figure 2) showed positive and significant slopes (slope = 0.56, *p* < 0.01) along the consistency line (*Y* = *X*), and the curvature did not reach a significant level (curvature = 0.03, *n.s.*), indicating that when FTP-FP were consistent and that followership increased as both scores rose. High FTP-high FP has stronger followership than low FTP-low FP, and H2 is supported. The curvature along the inconsistency line (*Y* = −*X*) reached a significant level (curvature = −0.21, *p* < 0.01), indicating that the followership increased with increasing FTP-FP consistency; H1 was verified. The slope was positive and significant (slope = 0.39, *p* < 0.01), indicating that in the case of inconsistency, the followership was significantly higher for high FTP-low FP than for low FTP-high FP, and H3 was verified.

#### 4.4.2. Mediating Effect Analysis

The FTP-FP consistency had a significant positive predictive effect on self-efficacy (*a*_1_ = 0.96, *p* < 0.01), and self-efficacy had a positive predictive effect on followership (*a*_2_ = 0.29, *p* < 0.01). As shown in Table 4, the indirect effect value of the FTP-FP consistency on followership through self-efficacy was 0.30, and the mediating effect test result (LLCI = 0.17, ULCI = 0.48) did not contain 0 at the 95% confidence interval, indicating a significant mediating effect. After controlling for self-efficacy, the effect of the FTP-FP consistency on followership remained significant (*a*_3_ = 0.69, *p* < 0.01), so further analysis was conducted to determine whether other mediating mechanisms existed. The calculation *a*_1_ × *a*_2_ × *a*_3_ = 0.21 > 0 indicates that there may be other complementary mediators that are consistent with the direction of the current mediating effect. In summary, self-efficacy partially mediated the effect of FTP-FP consistency on followership, and H4 was verified.

## 5. Discussion

In this study, we conceptualized follower identity by FTP-FP consistency, and examined self-efficacy as mediators between follower identity and followership. The results revealed that the relationship between follower-identity and followership is partially and significantly mediated by self-efficacy. The data analysis result reveals the support of all four hypotheses. The prediction of the first hypothesis regarding a positive relationship between employees’ FTP-FP consistency and followership is supported. That means employees who have higher FTP-FP consistency have stronger followership. This result is consistent with the previous hypothetical inference [31,32,33]. The prediction of the second hypothesis that “high FTP-high FP” individuals have stronger followership than “low FTP-low FP” individuals is also supported by our results, as well as by previous hypothetical inference [24,27,34,35]. The third hypothesis is also supported; the results reveal that “high FTP-low FP” employees have more substantial followership than “low FTP-high FP” individuals, as contended in earlier studies [29,36]. The mediation role of self-efficacy is partially and significantly substantiated in Hypothesis 4.

Previous studies have mainly explored the influence of factors, such as leadership style and leader expectations, on followership with a focus on leaders [46,47]. Leaders can transiently influence employees’ self-concept, but employees’ deeper cognitive schemas and values are difficult to be changed. Although some researchers have suggested that internal follower characteristics may impact followership, more relevant empirical studies must be conducted. This study indirectly measured the effect of follower identity on followership by examining the consistency of the “FTP-FP” in a combination of implicit and explicit ways. The study found that the consistency of FTP-FP facilitated the ideal schema into employees’ self-concept, enabling them to establish a more solid follower identity, which positively affected followership. In addition, employees’ actual following traits are highly aligned with the internal follower schema to maintain identity stability. Thus, there are differences in followership across different FTP-FP matching situations. The study responds to Uhl-Bien’s call for a “reverse perspective” to explore the influence of internal factors on followership [4].

Followership is not only formed due to passive influence by the leader and the environment but also the employees’ identity in playing the role of a follower also has a non-negligible role [48,49]. At the individual level, follower identity enables employees to clarify their followership and gives them a sense of competence; at the leader–member relationship level, relational identity enables employees integrated positive leader–member exchange into their self-concept, and they can get a sense of support. At the organizational level, organizational identity enhances the influence of group prototypes on individual behavior and gives employees a sense of belonging and security. The three levels of identity have distinct but closely related effects on individuals, and together they shape and influence employees’ followership performance. The study responds to the call of Epitropaki and other researchers to study the impact of follower identity from a broader range of perspectives [27].

Employees can achieve strong self-efficacy in FTP-FP congruence and exhibit excellent followership. This congruence positively influences employees’ self-efficacy by developing a series of successful experiences that result from their identity. Self-efficacy, in turn, affects followership by influencing employees’ perceptions and behaviors in the followership process. Further, the success experience brought by identity enhances self-efficacy assessment, and this confidence in self-efficacy, in turn, enhances employees’ followership performance, forming a virtuous circle. This study complements the leader perspective [42] and the leader–member relationship perspective by deepening the understanding of the mechanisms of follower self-efficacy formation and further validating the impact of self-efficacy on followership [50].

## 6. Conclusions

Followership is as crucial as leadership for organizational success. Significant efforts have been made by numerous researchers to examine how the transformational leadership [3,8,11], workplace authentic leadership [12], ethical leadership [6,13], servant leadership [14] and humble leadership [15] affects followership or other employees’ behaviors; however, enough attention has still not been given to the influence of internal factors of followers on followership from the followers’ perspective [16,17]. At the same time, only a few studies have examined the role of self-efficacy as mediators between follower identification and followership [18,19]. This study was grounded in the prediction of a follower identity approach to address the above-mentioned existing research gaps [4]. In this study, we conceptualized follower identity by FTP-FP consistency, and examined self-efficacy as mediators between follower identity and followership. The results revealed that the relationship between follower identity and followership is partially and significantly mediated by self-efficacy. The study responds to Uhl-Bien ‘s call for a “reverse perspective” to explore the influence of internal factors on followership [4], and deepens the understanding of how follower identity affects followership. For management practices, this study provides a new perspective on improving employee followership.

### 6.1. Implications

Followership, like leadership, is also subject to systematic learning [18]. The enhancement of followership should be devoted to improving employees’ understanding of themselves and fully mobilizing their initiative, which is conducive to promoting the sustainable development of the organization but also to enhancing employees’ happiness at work and in life. Accordingly, organizations should focus on the training and development of followership by developing a series of special training programs to improve employees’ self-insight, deepen their understanding of followership, and help them understand the value and importance of being good followers [51]. In addition, reflective training and prototype activation strategies can be used to guide employees to establish positive followership criteria [52].

Activation of followership enables employees to exhibit proactive followership behaviors under their identity criteria. Leaders can induce the follower schema in employees’ self-concept through transformational or charismatic leadership behaviors, such as personalized care, idealized influence, and inspirational motivation to help employees deepen their understanding of their strengths, mobilize and motivate their internal motivation, and give them a sense of value, meaning, and self-efficacy, thus effectively enhancing their followership performance [16].

More attention should be paid to the individual management of employees. For example, employees with consistent “FTP-FP” can be assigned more difficult and flexible tasks. For employees with inconsistent “FTP-FP”, it is recommended to encourage, not blame them, and to guide them step by step. For the organization, it should fully assist the HR system in order to support the performance of employees and adopt a flexible management approach according to the specific situation of employees [53,54]. For example, a high-commitment HRM approach is used for exceptional employees to give them more empowerment and motivation to evoke stronger followership.

### 6.2. Limitations and Future Research

First, identity establishment occurs in specific contexts, and employees seek the approval of the external social environment to strengthen their identity during the process of identity construction. Organizations can influence employees’ identity construction through sense-giving and sense-breaking [28]; therefore, future research can examine the effects of organizational culture and organizational climate on employee follower identity and followership. Second, followership is generated in the interaction between leaders and followers, and followership will also be constrained by the external environment. When follower schemas are not aligned with leadership expectations, such as passive followers working in a team with an empowered leader, they will experience more anxiety and dissatisfaction [32]. Future research could explore the effect of “leader-member” or “organization-member” matching on followership. Third, because there is no scale for follower identity, this study used the “FTP-FP” matching method for indirect measurement, which needs to be improved in terms of rigor and accuracy. More valid measures can be developed or applied for assessment in the future. Fourth, self-efficacy plays a partially mediating role between “FTP-FP” congruence and followership, and future studies can further explore other mediating variables in the model.

## Figures and Tables

**Figure 1 behavsci-13-00482-f001:**
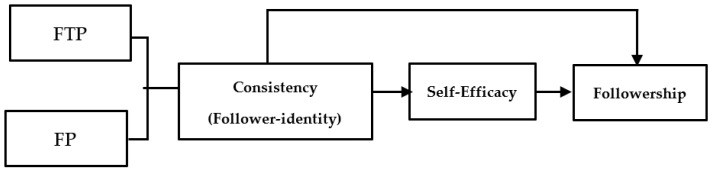
Research hypothesis model.

**Figure 2 behavsci-13-00482-f002:**
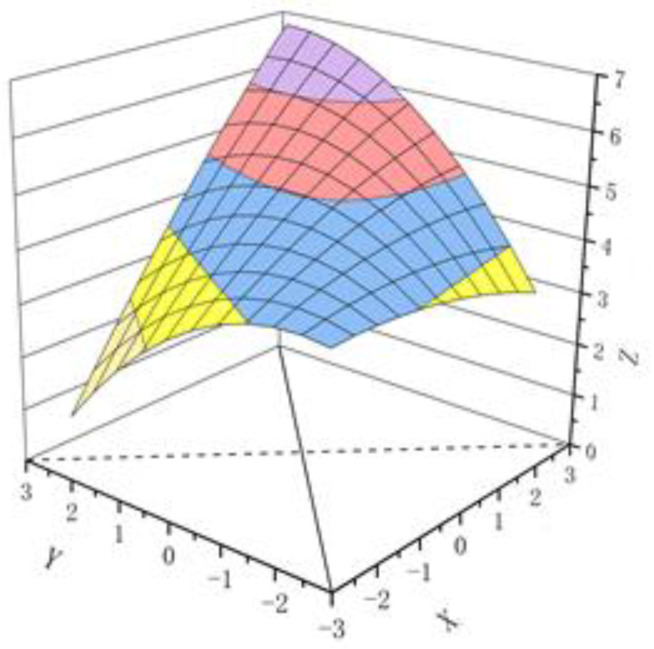
FTP-FP consistency and response surface analysis. Note: X = FTP (self-following trait perception), Y = FP (followership prototype), Z = FS (followership).

**Table 1 behavsci-13-00482-t001:** Results of validation factor analysis.

Models	χ^2^	*df*	Δχ^2^	RMESA	CFI	NFI	IFI	TLI
4 factors: FTP, FP, SE, FS	643.02	288	-	0.07	0.95	0.91	0.95	0.94
3 factors: FTP, FP, SE + FS	909.37	291	266.35	0.09	0.91	0.87	0.91	0.89
3 factors: FTP + FP, SE, FS	1468.98	291	825.96	0.12	0.82	0.79	0.83	0.79
2 Factor: FTP + FP + SE, FS	2098.61	293	1455.59	0.15	0.73	0.71	0.74	0.68
1 factor: FTP + FP + SE + FS	2326.12	294	1683.1	0.16	0.70	0.67	0.70	0.64

Note: N = 276, same as below; SE = self-efficacy, FS = followership.

**Table 2 behavsci-13-00482-t002:** Descriptive statistics.

	M	SD	1	2	3	4	5	6	7	8	9
1 Gender	1.47	0.50									
2 Age	2.15	10.03	−0.08								
3 Education	1.79	0.53	−0.01	−0.16 *							
4 Department	1.96	10.23	0.05	−0.13 *	0.04						
5 Years of service	2.00	10.14	−0.08	0.84 **	−0.07	−10					
6 FTP	6.81	10.64	0.09	0.38 **	−0.11	−0.08	0.35 **				
7 FP	7.67	10.40	0.07	0.29 **	−0.17 **	−0.05	0.19 **	0.26 **			
8 Self-efficacy	4.55	10.41	0.04	0.44 **	−0.11	−0.09	0.38 **	0.63 **	0.43 **		
9 Followership	5.23	10.43	−0.03	0.47 **	−14 *	−0.12	0.40 **	0.57 **	0.45 **	0.72 **	

Note: * indicates *p* < 0.05, ** indicates *p* < 0.01, two-tailed test, same below.

**Table 3 behavsci-13-00482-t003:** Results of polynomial regression and response surface analysis.

Variables	Followership		
	Model 1	Model 2	Model 3
Constant term	3.19	3.97	4.02
Control variables			
Gender	0.23	−0.01	0.04
Age	0.56 **	0.24 *	0.09
Education level	−0.11	0.05	0.05
Nature of the unit	−0.04	−0.01	0.05
Years of work	0.04	0.01	0.10
Independent variable			
FTP(*b*)_1_		0.43 **	0.47 **
FP(*b*)_2_		0.25 **	0.08
FTP squared (*b*)_3_			−0.02
FTP × FP (*b*)_4_			0.12 **
FP squared (*b*)_5_			−0.07*
R^2^	0.20	0.50	0.59
△R^2^		0.30 **	0.09 **
Response surface analysis			
Consistency line *Y* = *X*			
Slope (*b*_1_ + *b*)_2_			0.56 **
Curvature (*b*_3_ + *b*_4_ + *b*)_5_			0.03
Inconsistency line *Y* = −*X*			
Slope (*b*_1_ − *b*)_2_			0.39 **
Curvature (*b*_3_ − *b*_4_ + *b*)_5_			−0.21 **

**Table 4 behavsci-13-00482-t004:** Results of intermediate effect test.

Intermediary Model	Intermediate Effect Values (*a*_1_ × *a*)_2_	CI (95%)
FTP-FP consistency → self-efficacy →followership	0.96 × 0.32 = 0.31	[0.18, 0.47]

## Data Availability

Data supporting reported results are available from the authors on request.

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
