# Peer review of "The Effect of Follower Identity on Followership: The Mediating Role of Self-Efficacy"

_behavsci, 2023, doi:10.3390/bs13060482_

Round 1

Reviewer 1 Report

After careful consideration and review, we request major amendments to this manuscript before accepting it for publication.

Abstract

The abstract is relatively brief—a good informative abstract acts as a surrogate for the work itself. The researcher presents and explains the paper's main arguments, significant results, and evidence. An informative abstract includes the information found in a descriptive abstract [purpose, methods, scope]. However, it also consists of a judgment or comment about the study’s validity, reliability, or completeness, the results and conclusions of the research, and the author’s recommendations.

Introduction

The introduction does not adequately discuss how and why you planned to conduct this research, what the future benefits of this research will be to upcoming scholars, and what the findings of this study are. Make sure to:

1.       Establish an area to research by highlighting the importance of the topic, and/or making general statements about the topic, and/or presenting an overview of current research on the subject.

2.       Identify a research niche by opposing an existing assumption, and/or revealing a gap in existing research, and/or formulating a research question or problem, and/or continuing a disciplinary tradition.

3.       Place your research within the research niche by stating the intent of your study, outlining the key characteristics of your research, describing important results, and giving a brief overview of the structure of the paper.

Literature Review

The literature review misses important studies in this field. Additional review is needed, as well as the following studies should be included in this article:

·       Alghamdi, F. (2018). Ambidextrous leadership, ambidextrous employee, and the interaction between ambidextrous leadership and employee innovative performance. Journal of Innovation and Entrepreneurship7(1), 1-14.

·       Botsaris, C., & Vamvaka, V. (2016). Attitude toward entrepreneurship: Structure, prediction from behavioral beliefs, and relation to entrepreneurial intention. Journal of the Knowledge Economy7, 433-460.

·       Fachrunnisa, O., Siswanti, Y., El Qadri, Z. M., & Harjito, D. A. (2019). Empowering leadership and individual readiness to change: The role of people dimension and work method. Journal of the Knowledge Economy10, 1515-1535.

·       Soleas, E. K. (2020). Leader strategies for motivating innovation in individuals: a systematic review. Journal of Innovation and Entrepreneurship9, 1-28.

·       Vamvaka, V., Stoforos, C., Palaskas, T., & Botsaris, C. (2020). Attitude toward entrepreneurship, perceived behavioral control, and entrepreneurial intention: dimensionality, structural relationships, and gender differences. Journal of Innovation and Entrepreneurship9(1), 1-26.

Research Methodology

The descriptions of the research philosophy and design adopted in the study need to be sufficiently developed.

Research philosophy is a set of beliefs about collecting, analyzing, and using evidence concerning a phenomenon. Numerous research methods and philosophical frameworks are included under epistemology, which refers to what is known to be accurate, as opposed to doxology, which refers to what is thought to be true.

Research design is the blueprint for data collection, measurement, and analysis. The research design is the approach adopted to combine the numerous components of the study consistently and logically, thereby ensuring that they will successfully solve the research topic.

Elaborate further on the study’s sample and the strategies used to recruit participants.

Elaborate further on the study's data collection and analysis processes.

Overall, explain how you obtained and analyzed your findings because:

·       Readers need to know how the data was obtained because the method you chose affects the results and, by extension, how you interpreted their significance in the discussion section of your paper.

·       The methodology is crucial for any branch of scholarship because an unreliable method produces unreliable results and, consequently, undermines the value of your analysis of the findings.

·       There are various methods you can choose to investigate a research problem. The methodology section of your paper should clearly articulate why you have chosen a particular procedure or technique.

·       The reader wants to know whether the data was collected or generated consistently with accepted practice in the field of study.

·       The method must be appropriate to fulfill the study’s overall aims.

·       The methodology should discuss the anticipated problems and the steps you took to prevent them from occurring. For any problems that arise, you must describe how they were minimized or why they do not impact your interpretation of the findings in any meaningful way.

·       It is essential to provide sufficient information to allow other researchers to adopt or replicate your methodology. This information is critical when a new method has been developed, or innovative use of an existing method is utilized.

Analysis of Findings

Although the Results section provides a detailed description of the data collected, there needs to be a more critical synthesis and comparison of the findings in the analysis of the results.

Discussion

The Discussion section is missing. The purpose of the discussion section is to interpret and describe the significance of your findings in relation to what was already known about the research problem being investigated and explain any new understanding or insights that emerged from your research. The discussion connects to the introduction through the research questions or hypotheses and the literature you reviewed. The Discussion should include a critical synthesis and comparison of the data with the literature. The discussion clearly explains how your study advanced the reader’s understanding of the research problem from where you left them at the end of your review of prior research. The content of the discussion section of your paper should include:

·       Explanation of results: Comment on whether the results were expected for each set of findings; go into greater depth to explain unexpected or incredibly profound findings. If appropriate, note any unusual or unanticipated patterns or trends that emerged from your results and explain their meaning concerning the research problem.

·       References to previous research: Either compare your results with the findings from other studies or use the studies to support a claim. This can include re-visiting key sources already cited in your literature review section or saving them from citing later in the discussion section if they are more important to compare with your results instead of being a part of the general literature review of prior research used to provide context and background information.

·       Deduction: A claim for how the results can be applied more generally. For example, describing lessons learned, proposing recommendations that can help improve a situation, or highlighting best practices.

·       Hypothesis: A more general claim or possible conclusion arising from the results [which may be proved or disproved in subsequent research]. This can be framed as new research questions that emerged from your analysis.

Conclusion

The Conclusion is underdeveloped and does not adequately discuss the theoretical and managerial implications of the study. Summarize your thoughts and convey the larger significance of your research. Identify and discuss how a gap in the literature has been addressed and demonstrate the importance of your ideas. Introduce possible new or expanded ways of thinking about the research problem.

Also, state the ideas for future research in the conclusion. Make sure you create 3 subsections in the Conclusion: 1) Theoretical implications, 2) Managerial Implications, and 3) Ideas for Future Research.

You may wish to study published articles that examine perspectives on this topic, which will give you an idea of how you must revise your article.

Make sure to proofread the manuscript before it is resubmitted to the journal. Please go through the journal’s guidelines thoroughly and revise the paper accordingly.

Reviewer 2 Report

The authors aimed to investigate the, “The effect of follower identity on followership: The mediating role of self-efficacy”.  Overall, the manuscript is well written in all sections of the manuscript, followed by sound methods with results from wide study settings. The study is meaningful in the current situation and context. 

However, I would like to provide the following comments and suggestions being considered before being accepted for publication. 

Authors, should clarify why self-efficacy is used as mediator? In the Figure 1 of the paper heading’” Research hypothesis model” it is not clear that the self-efficacy is a mediator. Could authors re-write and clarify in the text as well so that the novice readers can also understand. In the Abstract, I suggest authors mention the total number of study subjects as well in addition to study design. The results should be a bit more clear to the reader. Conclusion should be added in the abstract. 

Keywords are usually written alphabetically. Could authors first describe the outcome variables in detail and then for covariates? Results are expected to be a bit more clear to the reader. Could authors re-write the results section of the paper. In the discussion section, authors should highlight the strengths of the study while limitations should be extended more precisely noting down biases associated with.

Reviewer 3 Report

Thank you, the authors, to pick up a very relevant topic, that is, “The effect of follower identity on followership: The mediating role of self-efficacy”. I have the following investigations on the article:

1.     Comments on Introduction

a.      The authors should include the justification of the research, it’s missing.

b.     The authors should figure out the research gap exactly.

c.      The authors must include the contribution to this current study.

d.     The authors should specify the next chapters of the article.

2.     Comments on Literature review

a.      The current literature review is very week. The authors must include more studies and updated literature considering current studies like 2017-2023.

b.     The literature gap is missing here. The authors must find out the literature gap.

c.      The authors should include the application of the used theory to theoretical background on the literature review section.

d.     The authors should ignore some recent studies. The followings can be included:

·        Bantha, T. and Sahni, S.P., 2021. The relation of servant leadership with followers' organizational citizenship behaviour (OCB): mediating role of generalized self-efficacy (GSE) and organization–based self-esteem (OBSE). Industrial and Commercial Training53(4), pp.331-342.

·        Al Hawamdeh, N., 2022. Does humble leadership mitigate employees’ knowledge-hiding behaviour? The mediating role of employees’ self-efficacy and trust in their leader. Journal of Knowledge Management, (ahead-of-print).

·        Salanova, M., Rodríguez-Sánchez, A.M. and Nielsen, K., 2022. The impact of group efficacy beliefs and transformational leadership on followers’ self-efficacy: A multilevel-longitudinal study. Current Psychology41(4), pp.2024-2033.

·        Ashfaq, F., Abid, G. and Ilyas, S., 2021. Impact of ethical leadership on employee engagement: role of self-efficacy and organizational commitment. European Journal of Investigation in Health, Psychology and Education11(3), pp.962-974.

3.      Comments on Methodology

a.      The authors should use standard methodology. The data collection process is suspicious. The sample technique is missing. The justification to select the sample size must be given, that is missing here. Justification for the selected data analysis plan must be specified here.

b.      The authors can use Structural Equation Modeling (SEM) to analyze the data.

4.      Comments on Results

a.      The developed model is interesting. But inclusive of justification for multicollinearity issues, convergent validity, divergent validity, internal consistency, and model fitness will enhance the quality of the findings as well as the article.

b.      The results should be connected to the previous studies.

5.      Comments on Discussion and conclusions

a.      The discussion part is ok.

b.      The conclusion sections should have and update the theoretical implications, managerial implications, limitations, and future study directions.

Reviewer 4 Report

This paper investigated the 11 influence of followers' perceived self-following traits (FTP) and followership prototype (FP) on fol-12 lowership, as well as the mediating role of self-efficacy. Polynomial regression, response surface 13 analysis, and mediation effect tests were conducted on survey data from 276 employees, and the 14 results indicated that (1) the more consistent FTP-FP, the stronger the followership; (2) compared to 15 'low FTP-low FP,' employees with 'high FTP-high FP' had stronger followership; (3) employees with 16 'high FTP-low FP' had stronger followership than 'low FTP-high FP'; (4) self-efficacy played a me-17 diating role between FTP-FP consistency and followership.

Three issues need to be addressed in the revised manuscript:

1. a comprehensive literature survey on the topic

2. identify the knowledge gap based on the comprehensive  literature survey 

3. Describe the research's contribution to methodology. 

Round 2

Reviewer 3 Report

Thank you for replying the previous comments. There are further scopes to improve the methodology of the research paper.

Reviewer 4 Report

Authors addressed comments approportely. Good luck!
